# Brain Death and Its Prediction in Out-of-Hospital Cardiac Arrest Patients Treated with Targeted Temperature Management

**DOI:** 10.3390/diagnostics12051190

**Published:** 2022-05-10

**Authors:** Hwan Song, Sang Hoon Oh, Hye Rim Woo

**Affiliations:** 1Department of Emergency Medicine, St. Vincent’s Hospital, College of Medicine, The Catholic University of Korea, Seoul 06591, Korea; cmcmdsong@gmail.com; 2Department of Emergency Medicine, Seoul St. Mary’s Hospital, College of Medicine, The Catholic University of Korea, Seoul 06591, Korea; 3Department of Emergency Medicine, Bucheon St. Mary’s Hospital, College of Medicine, The Catholic University of Korea, Seoul 06591, Korea; 19215@naver.com

**Keywords:** heart arrest, induced hypothermia, prognostication, brain death, organ transplantation

## Abstract

Evolution toward brain death (BD) in out-of-hospital cardiac arrest patients with targeted temperature management (TTM) provides opportunities for organ donation. However, knowledge regarding BD in these patients is limited. We retrospectively analyzed the TTM registry of one hospital where life-sustaining therapy was not withdrawn. In-hospital death patients were categorized into BD and non-BD groups. We explored the process of evolution toward BD and its predictors by comparing the serial measurements of clinical variables and the results of various prognostic tests between the two groups. Of the 121 patients who died before hospital discharge, 19 patients (15.7%) developed BD at a median of 6 (interquartile range, 5.0–7.0) days after cardiac arrest. Four patients with pupillary light reflexes at 48 h eventually developed BD. The area under the curves of the gray-to-white matter ratio (GWR) on early brain computed tomography images and the level of S100 calcium-binding protein B (S100B) at 72 h were 0.67 (95% CI, 0.55–0.77) and 0.70 (95% CI, 0.55–0.83), respectively. In conclusion, approximately one-sixth of all in-hospital deaths were diagnosed with BD at a median of 6 days after cardiac arrest. The use of GWR and serial S100B measurements may help to screen potential BD.

## 1. Introduction

Recent advances in postcardiac arrest care, such as targeted temperature management (TTM) and emergency coronary interventions, have improved neurological outcomes after out-of-hospital cardiac arrest (OHCA) [1,2]. However, patient mortality remains high, and approximately two-thirds of patients admitted to the hospital die before hospital discharge [3,4]. Most of these deaths are due to hypoxic brain injury with or without active withdrawal of life-sustaining treatment (WLST) based on prognostication with a poor neurological outcome [5,6]. However, a small portion of these patients show a permanent absence of all cerebral and brainstem functions from massive cerebral edema [7,8], resulting in brain death (BD) [9].

Postcardiac arrest patients make up an increasing proportion of the organ donor pool [10], and these patients provide a mean of 3.9 or 2.9 organs per donor [10,11]. Moreover, the organ function of brain-dead donors after cardiac arrest is similar to that of organ donors who died from other causes [12]. The international guidelines of cardiopulmonary resuscitation (CPR) recommend that all patients who are resuscitated from cardiac arrest and subsequently progress to death or BD should be evaluated as potential organ donors [13,14]. However, knowledge of the evolution of these patients to BD is very limited [9]. Criteria for the diagnosis of BD are significantly heterogeneous among studies, and WLST after failure to improve neurologically might affect donation prevalence after diagnosis of BD [15]. Previous studies have reported some early factors related to the evolution of BD [16,17,18,19]. However, it is unclear whether the prognostication tools recommended by international guidelines for post-resuscitation care to predict neurological outcomes can also predict the evolution toward BD [14]. Moreover, this analysis should be conducted on non-survivors. In South Korea, BD was diagnosed when both clinical and confirmatory tests were fulfilled [20], and decisions regarding the termination of life support after OHCA were not allowed until 2017. Therefore, if BD was not confirmed, they were continuously treated without WLST.

In this study, we analyzed clinical variables, including neurological examination and hemodynamic state, during the 5 days after return of spontaneous circulation (ROSC) and various prognostic tests among in-hospital deaths. The primary aim of the current study was to evaluate the process of evolution toward BD in OHCA patients with TTM. The secondary aim was to identify prognostic tools related to evolution toward BD.

## 2. Materials and Methods

### 2.1. Study Design and Patients

This study retrospectively analyzed the TTM registry data at one tertiary hospital in South Korea. We included consecutive patients who were at least 18 years old and were treated with TTM after OHCA between 2012 and 2018. The study protocol was approved by the Institutional Review Board of the Seoul St. Mary’s hospital (KC20RISI0376) and was performed in accordance with the ethical guidelines of the Declaration of Helsinki. Informed consent was waived because of the retrospective nature of the study.

### 2.2. Postcardiac Arrest Care

During the study period, all adult OHCA patients who were comatose after ROSC were eligible for TTM, usually at 33 °C for 24 h [14]. After the target temperature of 33 °C was maintained for 24 h, controlled rewarming at a rate of 0.25 °C/h was performed until the temperature reached 36.5 °C. Patients received a combination of midazolam and rocuronium during TTM, and this treatment was reduced during rewarming and discontinued as soon as the central temperature reached 35 °C. Considering the potential risks of TTM at 33 °C, some patients were treated with a target temperature of 36 °C. The use of extracorporeal membrane oxygenation (ECMO) was reserved for cases of cardiac shock that was refractory to vasopressor medications for hemodynamic support according to the discretion of the attending physicians and was not used for extracorporeal CPR. In order to monitor the hemodynamic state, the level of vasopressor support was assessed daily by the cardiovascular subscore of the Sequential Organ Failure Assessment (SOFA-C) [21].

A standardized approach for prognostication was applied to all patients. Comatose patients routinely underwent nonenhanced brain computed tomography (CT) scans immediately after ROSC. Neurological examinations were serially performed according to standard practices. In all participants, we evaluated whether pupillary light reflex (PLR) and spontaneous eye opening were bilaterally present or absent and the Glasgow coma scale (GCS) motor grade for 5 days. Measurements of serum neuron-specific enolase (NSE) and S100 calcium-binding protein B (S100B) were obtained immediately after ROSC and repeated 24, 48, and 72 h later. The serum was analyzed with Roche Elecsys NSE and S100 reagents (Roche Diagnostics, Mannheim, Germany). If the serum showed significant hemolysis, the results were discarded. Somatosensory evoked potential (SSEP) measurements and diffusion-weighted imaging (DWI) were usually performed after completing rewarming. Because South Korean law prohibited physicians from WLST, even if poor outcomes were predicted, we continuously treated patients, and WLST was not considered for any participant.

### 2.3. Analyses of Outcome Predictors

On brain CT, the gray-to-white matter ratio (GWR) in the basal ganglia was measured using the regions of interest function in Hounsfield units according to the methods described in previous studies (Appendix A) [22]. DWI findings were categorized into three patterns on the basis of the diffusion-restriction lesions of the brain: (1) no diffusion restriction, (2) isolated cerebral cortex or deep gray matter, and (3) multifocal and global diffusion restriction [23]. In the analysis of the SSEP, we focused on whether the N20 responses were clearly discernible to categorize as present or absent [24].

### 2.4. Declaration of BD and Organ Donation (OD)

BD determinations were conducted twice by at least two specialist physicians. The patient should be off sedation or neuromuscular paralysis and complicating medical conditions that may confound the clinical assessment should also be corrected for an accurate evaluation. The examination must demonstrate deep unresponsive coma with the absence of cerebral and all brainstem functions on neurological examinations, and apnea tests (or brain blood flow studies as ancillary tests) should be completed [25]. Because South Korea mandates the routine use of confirmatory tests to supplement the clinical examination, electrocerebral silence during 30 min of recording should be established [20]. Once the assessment is complete, a follow-up evaluation is mandatory after 6 h.

### 2.5. Statistical Analysis

Categorical variables are expressed as the number and percentage, and continuous variables are expressed as the mean and standard deviation or the median and interquartile range (IQR) for normally distributed variables. Comparisons of differences in characteristics and other variables between the BD and non-BD groups were performed using chi-square tests or Fisher’s exact tests for categorical variables and Student’s *t*-test or the Mann–Whitney U test for continuous variables. Serial results between the two groups were analyzed as absolute differences in proportions with 95% confidence intervals (CIs). Receiver operating characteristic (ROC) curves were generated for continuous predictors and combined model using logistic regression analysis, and the predictive accuracy was determined by the area under the ROC curve (AUC), the cutoff values, and the sensitivities and specificities with the 95% CI. All statistical analyses were performed using IBM SPSS version 24 software (IBM, Armonk, NY, USA) and the Medcalc program (Medcalc Software, Mariakerke, Belgium). All p-values were two-tailed, and *p* < 0.05 was considered significant.

## 3. Results

### 3.1. Baseline Characteristics of Participants

During the study period, a total of 251 adult OHCA patients were treated with TTM, and 121 patients died before hospital discharge (Figure 1). Among these, 19 patients were diagnosed with BD, and all underwent organ retrieval. The BD and non-BD (in-hospital death from other causes) groups showed distinct baseline characteristics, as described in Table 1. The BD group was significantly younger and had lower incidences of witnessed arrest and comorbidities, such as hypertension, than the non-BD group (all *p* < 0.05). In the BD group, asphyxia (68.4%) was the main cause of cardiac arrest and was followed by cardiac and other medical causes (both, 15.3%), which was significantly different from the non-BD group (*p* < 0.001). Finally, 72 organs were donated from these patients (33 kidneys, 18 livers, 8 hearts, 7 pancreases, and 6 sets of lungs) (Appendix A).

### 3.2. Clinical Course of Patients with BD

The median time to a legal BD diagnosis was 6.0 (IQR, 5.0–7.0) days. This duration was similar to that of death from other causes (median 4.5 [IQR, 2.0–8.3] days, *p* = 0.158). The shortest time to diagnosis of BD was 3 days after ROSC (n = 1), 4 days (n = 3), 5 days (n = 5) and 6 to 12 days (n = 10) (Figure 1). In-hospital death patients usually showed an absence of eye response and a GCS motor grade ≤ 2 regardless of final outcomes, and both clinical examinations were not different between the BD and non-BD groups during the observation period (Figure 2A,B). On the other hand, the incidence of absence of PLR significantly differed between the two groups over time (3 days, *p* = 0.005 and 5 days, *p* = 0.015) (Figure 2C). We also found that four patients who had PLR at 24 h and 48 h after ROSC finally progressed to BD. Among these patients, one still maintained PLR 5 days after ROSC.

In approximately half of the BD patients, high doses of vasopressors were continuously required during monitoring, while the hemodynamics of the non-BD group improved over time regardless of whether the patients who deteriorated died (Figure 2D). Five days after ROSC, the number of patients without vasopressor support increased in the non-BD group, and the proportions of subjects with high doses of vasopressor support (SOFA-C = 4) were likely higher in the BD group than in the non-BD group, although these differences did not reach statistical significance (*p* = 0.075).

### 3.3. Results of other Prognostic Tests

Ninety-six subjects had a brain CT scan. The median ROSC-to-CT interval was 17.0 (IQR, 10.3–29.8) min, and the mean GWR was significantly reduced in BD patients compared to non-BD subjects (1.12 ± 0.08 vs. 1.18 ± 0.07, *p* = 0.003) (Table 2). We used the ROC curve to assess GWR as a predictor of BD (AUC = 0.67; 95% CI, 0.55–0.77) (Figure 3A). A GWR value ≤ 1.06 predicted BD with a sensitivity of 27.8% (95% CI, 9.7–53.5) and a specificity of 96.2% (95% CI, 89.2–99.2).

Sixty-five subjects had an SSEP measurement at a median of 45.0 (IQR, 30.1–61.8) h after ROSC. While all BD patients showed an absent N20, eight of 54 SSEPs in the non-BD group had a present N20. However, the SSEP pattern was not significantly different between the two groups (*p* = 0.104).

Forty-three subjects received DWI scans at a median of 65.5 (IQR, 56.1–81.5) h. All BD patients and most non-BD patients showed multifocal or global diffusion restriction on DWI images, and there was no significant difference between the two groups (*p* = 0.139).

The BD group had higher peak levels of NSE than the non-BD group (170.6 [IQR, 91.2–210.0] ng/mL vs. 67.6 [IQR, 33.5–200.0] ng/mL, *p* = 0.011), and ROC analysis revealed that the peak level of NSE had an AUC for BD of 0.54 [95% CI, 0.38–0.69]. Median NSE levels steadily increased as time elapsed, and changes in NSE over time were similar between the two groups (Figure 4). On the other hand, the peak levels of S100B were not different between the two groups (7.3 [IQR, 3.0–21.1] ng/mL vs. 5.2 [IQR, 2.6–13.5] ng/mL, *p* = 0.288). Median S100B levels decreased in most patients in the non-BD group 3 days after ROSC. In contrast, those in the BD group abruptly increased at 3 days and were significantly higher than those in the non-BD group (10.4 [IQR, 0.4–16.5] ng/mL vs. 0.9 [IQR, 0.3–4.3] ng/mL, *p* = 0.040). The AUC of the levels of S100B at 72 h for BD was 0.70 (95% CI, 0.55–0.83) (Figure 3B), and a S100B > 9.09 ng/mL predicted BD with a sensitivity of 58.4% (95% CI, 27.7–84.8) and a specificity of 87.5% (95% CI, 71.0–96.5). The AUC of the combination of the GWR and S100B at 72 h was 0.75 (95% CI, 0.58–0.88) (Figure 3C).

Details of prognostic tests in BD patients are provided in Appendix A.

## 4. Discussion

In our study, 7.6% of patients with TTM after OHCA progressed to BD at a median of 6 days after ROSC and ultimately donated organs. They represented approximately one-sixth of all in-hospital deaths. Four patients who had PLR at 24 h and 48 h after ROSC finally progressed to BD. Among various outcome predictors, early brain CT findings and serial measurement of serum S100B, especially the results at 72 h, had a potential role in the identification of patients who could evolve toward BD.

Our prevalence of BD and OD is likely to be higher than those in a recent systematic review published in 2016 [9], in which the estimated pooled prevalence of BD was 5.4% in patients successfully resuscitated with conventional CPR, corresponding to 8.3% of all in-hospital deaths, and the rate of OD was reported to be less than 50% among all BD cases. However, the results should be cautiously interpreted in the context of several biases. It is difficult to measure the real prevalence of BD after successful resuscitation from OHCA. Some patients have neurodiagnostic results that were substantively similar to those in the BD group but inevitably end up classified as non-BD because they suffered rearrest or multisystem organ failure. This bias is present in most studies and is very difficult to address statistically. Furthermore, the diagnostic criteria for BD were heterogeneous among studies or countries [25]. In South Korea, to be diagnosed with BD and be recognized as legally dead, the patient’s family must consent to the evaluation process for purposes of possible OD [26]. Therefore, an undiagnosed BD could be underreported, and the differences in some baseline characteristics between the two groups may simply reflect the criteria used to separate eligible donors from noneligible donors, although these biases would not influence the incidence of OD. 

Several factors may explain the difference in our prevalence of BD from other studies. Standard care, including TTM, may lead to improvements in cardiovascular and other organ functions. Since cardiac arrest damages other organs along with the brain [27,28], BD is pronounced less frequently than acute brain damage of different vascular and traumatic etiologies [29]. However, performing TTM can impact central hemodynamics, as both the cardiac index and heart rate are decreased, and the need for vasopressors is increased [30,31], and paradoxically, secondary systemic insults could aggravate the brain and vital organ injury [32]. Although the proportions of SOFA-Cs were not different between the two groups, approximately more than half of the patients in the BD group continuously needed high doses of vasopressor support, and several BD patients were confirmed to have worsened over time upon neurological examination. However, to understand the process of BD evolution in these patients, various aspects should be considered, and it is still unknown how the patient’s hemodynamics are related to the development of BD.

It also deserves further mention that many nonshockable rhythms and noncardiac OHCAs were included in this analysis [18]. Although recent landmark trials have shown inconclusive effects [1,33], international CPR guidelines recommend that TTM should be considered in these patients [13,14]. Nevertheless, some facilities do not actively perform TTM for these patients [34]. Based on the benefit (“doing good”) of resuscitation ethical principles, our data showed that TTM in these patients not only could improve prognosis in some patients but also serve as a bridge to OD. Various factors might play a secondary brain injury [35,36], and continuing intensive care may provide opportunities for progression toward BD, which also explains the extended period before BD diagnosis. Since the death of patients who have an opportunity to evolve into BD can be exacerbated by the substantial delay of BD pronouncement, some experts have suggested that BD can be suspected earlier and confirmed even before TTM or rewarming [9]. There is concern that these patients could also sometimes survive for extended periods of time, in which continued treatment is no longer proportional to the potential benefits of that treatment. However, the cooling or medication used during TTM delays the initiation or completion of BD protocols, acting as confounding factors similar to those in formal prognostication. This seems to conflict with ethical principles. Therefore, it is important to screen patients who will evolve toward BD within the time interval of the current prognostication practice, especially among patients who have a poor outcome (in-hospital death) [13,14].

Recently, BD after cardiac arrest score showed good performance, in which natremia at 24 h and neurological cause of OHCA were the two robust factors [37]. However, little is known about which prognostication tools in the TTM cohort could predict the evolution toward BD. Interestingly, our results for the GWR cutoff were similar to those reported by Scarpino et al. [17], who had used a similar approach and showed that combining early SSEP (<24 h) and CT scan data predicted BD after cardiac arrest. However, we failed to determine whether the SSEP has discriminative power for BD from death from other causes. Our BD patients may have had marked brain injury because they were diagnosed by confirmation tests, but we believed it was difficult to distinguish between the two groups because our non-BD cohort also had poor resuscitation variables. It is still unclear whether body temperature contributes to the difference between the two results.

Our results revealed that a significant number of BDs can be diagnosed after completing TTM and multimodal prognostication recommended by the CPR guidelines [13,14]. The serum level of S100B at 72 h was abruptly elevated in BD patients, consistent with previous studies reporting that S100B measurements were significantly higher in the group with confirmed BD than in the other group [38,39], although these studies did not focus on postcardiac arrest or in-hospital death. S100B has the highest sensitivity for poor neurological outcomes during the first 24 h after ROSC because of its very short half-life and usually declines over the next 48 h [40]. Further increases are probably due to persistent brain damage, leading to BD. Considering that neurologic prognostication is usually performed 72 h after ROSC, or later in the TTM era, these laboratory values could also provide information to determine the likelihood that a patient will eventually progress toward BD if intensive care is continued.

This study had several limitations. First, as mentioned above, a major limitation was the possibility of misclassification due to the inability to confirm BD in patients due to death from other causes or nonconsent for OD. Second, the number of BD patients might have been too small to draw conclusions and may restrict the generalization of our results. Thus, larger, prospective studies are warranted. Third, as hemolysis leads to falsely elevated NSE and S100B values, we discarded samples with visible hemolysis. However, the absence of detectable hemolysis may not completely rule out the absence of NSE and S100B from extracerebral origins, especially in patients with ECMO support. Fourth, although WLST was not considered, do-not-resuscitate orders were legal and socially acceptable. Therefore, the results of prognostic tests could potentially have influenced decisions regarding withholding advanced treatment. Lastly, our retrospective registry did not include potentially related factors, such as diabetes insipidus with hypernatremia, which is usually an early event preceding BD [37,41].

## 5. Conclusions

Our study showed that 19 of 121 patients who died before hospital discharge progressed toward BD at a median of 6 days after ROSC and had organs retrieved. Approximately one-fifth of them deteriorated after rewarming and progressed toward BD. We found that the early GWR and the serum S100B level at 72 h were reliable predictors for the evolution of BD among in-hospital deaths. Our results could help physicians organize timely screenings and identify potential BD among these patients.

## Figures and Tables

**Figure 1 diagnostics-12-01190-f001:**
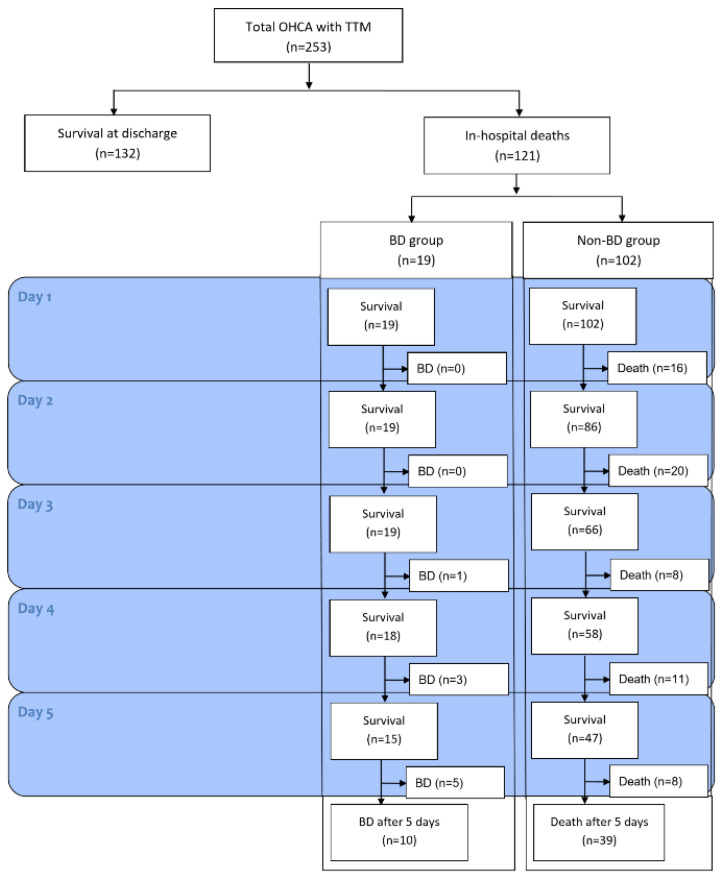
Flow chart for inclusion of patients in the study. OHCA, out-of-hospital cardiac arrest; TTM, targeted temperature management; BD, brain death.

**Figure 2 diagnostics-12-01190-f002:**
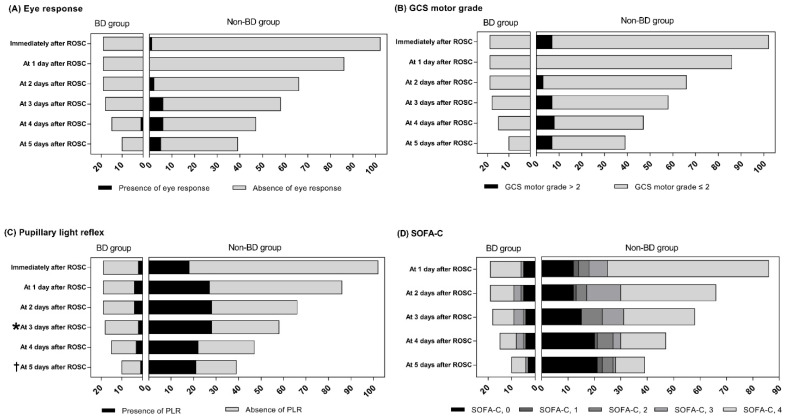
Distributions of in-hospital death patients during 5 days after ROSC between BD and non-BD patients according to Eye response (**A**), GCS motor grade (**B**), pupillary light reflex (**C**) and SOFA-C (**D**). * *p* = 0.005 and † *p* = 0.015. BD, brain death; GCS, Glasgow coma scale; SOFA-C, cardiovascular Sequential Organ Failure Assessment score.

**Figure 3 diagnostics-12-01190-f003:**
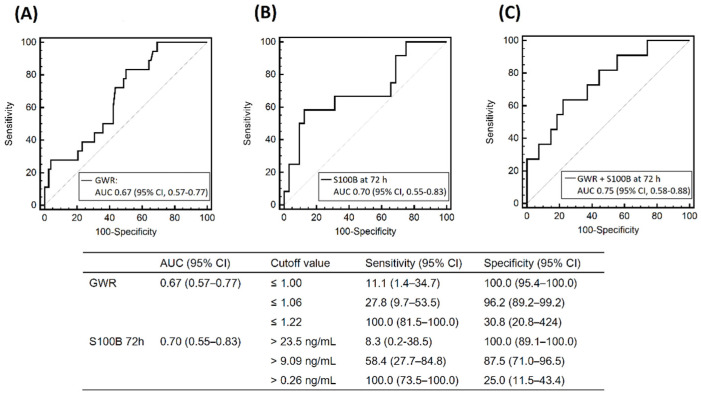
The receiver operating characteristic curves for evolution toward brain death after out-of-hospital cardiac arrest. (**A**) The AUC for GWR. (**B**) The AUC for the serum level of S100B protein at 72 h after cardiac arrest. (**C**) The AUC for the combined model with GWR and S100B at 72 h after cardiac arrest. AUC, area under the curve; CI, confidence interval; GWR, gray-to-white matter ratio at basal ganglia; S100B, S100 calcium-binding protein B.

**Figure 4 diagnostics-12-01190-f004:**
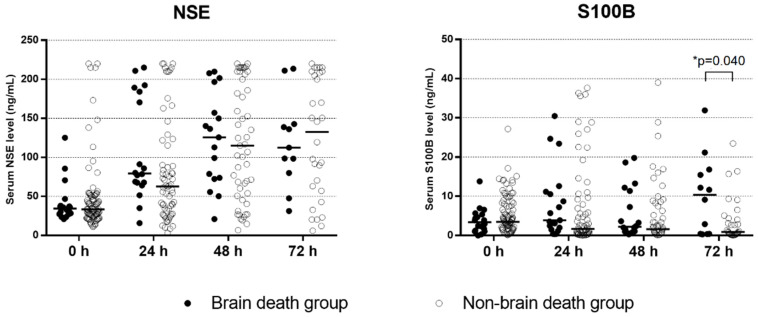
Serial serum NSE and S100B levels in the brain death group and the non-brain death group. Bars represent medians. * *p* = 0.040 using the Mann–Whitney U test. NSE, neuron-specific enolase; S100B, S100 calcium-binding protein B.

**Table 1 diagnostics-12-01190-t001:** Characteristics of the included patients.

	Survivors (n = 130)	In-Hospital Deaths (n = 121)	*p*-Value	In-Hospital Deaths (n = 121)		
				BD (n = 19)	Non-BD (n = 102)	*p*-Value
Male, n (%)	90 (69.2)	91 (75.2)	0.291	13 (68.4)	78 (76.5)	0.563
Age, years, mean ± SD or median (IQR)	52.9 ± 14.5	60.7 ± 17.4	<0.001	45.0 (33.0–54.0)	68.9 (54.0–75.5)	<0.001
Comorbidities, n (%)						
Hypertension	38 (29.2)	51 (42.1)	0.033	0 (0.0)	51 (50.0)	<0.001
Diabetes mellitus	25 (19.2)	38 (31.4)	0.026	3 (15.8)	35 (34.3)	0.177
COPD	5 (3.8)	17 (14.0)	0.004	1 (5.3)	16 (15.7)	0.305
Ischemic heart disease	19 (14.6)	14 (11.6)	0.476	0 (0.0)	14 (13.7)	0.123
Chronic renal failure	6 (4.6)	13 (10.7)	0.067	1 (5.3)	12 (11.8)	0.690
Presumed cause, n (%)			<0.001			<0.001
Cardiac	102 (78.5)	56 (46.3)		3 (15.8)	53 (52.0)	
Submersion	0 (0.0)	5 (4.1)		0 (0.0)	5 (4.9)	
Drug	0 (0.0)	2 (1.7)		0 (0.0)	2 (2.0)	
Asphyxia	20 (15.4)	32 (26.4)		13 (68.4)	19 (18.6)	
Other-medical	8 (6.2)	26 (21.5)		3 (15.8)	23 (22.5)	
Witnessed, n (%)	99 (76.2)	74 (61.2)	0.010	6 (31.6)	68 (66.7)	0.009
Initial shockable rhythm, n (%)	64 (49.2)	26 (21.5)	<0.001	2 (10.5)	24 (23.5)	0.360
Bystander CPR, n (%)	86 (66.2)	72 (59.5)	0.276	12 (63.2)	60 (58.8)	0.803
Arrest time *, min, mean ± SD	27.4 ± 15.9	40.9 ± 22.2	<0.001	41.0 (33.0–54.0)	39.0 (28.0–51.0)	0.117
LOS, day, median (IQR)	13.0 (8.3–21.0)	5.0 (2.2–8.0)	<0.001	6.0 (5.0–7.0)	4.5 (2.0–8.3)	0.158

* Arrest time was defined as the interval form arrest to return of spontaneous circulation. BD, brain death; SD, standard deviation; IQR, interquartile range; COPD, chronic obstructive pulmonary disease; CPR, cardiopulmonary resuscitation; ROSC, return of spontaneous circulation; LOS, length of hospital stay.

**Table 2 diagnostics-12-01190-t002:** Contingency table depicting the relationship between various outcome predictors and the occurrence of BD.

	BD (n = 19)	Non-BD (n = 102)	*p*-Value
GWR at basal ganglia, n = 96	1.12 ± 0.08, n = 18	1.18 ± 0.07, n = 78	0.003
Pupillary light reflex at 3 days, n = 76			0.005
Present reflex	2 (11.1)	28 (48.3)	
Absent reflex	16 (88.9)	30 (51.7)	
NSE at 72 h, ng/mL, n = 41	112.4 (IQR, 79.8–142.7), n = 11	132.5 (IQR, 53.5–208.8), n = 30	0.860
S100B at 72 h, ng/mL, n = 44	10.4 (IQR, 0.4–16.5), n = 12	0.9 (IQR, 0.3–4.3), n = 32	0.040
Diffusion-weighted imaging, n = 43			0.307
No lesion	0 (0.0)	2 (7.1)	
Isolated cortex or deep gray matter lesion	0 (0.0)	2 (7.1)	
Multifocal or global lesion	15 (100.0)	24 (85.7)	
Somatosensory evoked potential, n = 65			0.104
Present N20	0 (0.0)	12 (23.1)	
Absent N20	13 (0.0)	40 (76.9)	

BD, brain death; GWR, gray-to-white matter ratio at basal ganglia; NSE, neuron-specific enolase; IQR, interquartile range; S100B, S100 calcium-binding protein B.

## Data Availability

The data presented in this study are available on request from the corresponding author. The data are not publicly available due to legal restrictions.

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
