# Peer review of "Brain Death and Its Prediction in Out-of-Hospital Cardiac Arrest Patients Treated with Targeted Temperature Management"

_diagnostics, 2022, doi:10.3390/diagnostics12051190_

Round 1

Reviewer 1 Report

Dear Authors.

 Thank you for the opportunity to review this submission. As a standard practice, I have thoroughly read this submission before coming to this conclusion. It was interesting to predict brain death after post-cardiac arrest care. Although this manuscript is well-written, it has a few limitations that need to be addressed before its acceptance.

  1. Interestingly, the median age of the BD group was 45 years which is significantly younger than that of the non-BD group. Is it because of socio-cultural differences?
  2. The authors used an “eye-opening/response” as a tool for predicting BD. However, its definition was not clear. Please clarify whether it is spontaneous or reactive to voice or pain. It can lead a confusion to readers. Please clarify the definition of “eye opening/response”
  3. ECMO induced hemolysis is widely accepted as one of the major complication of one. Therefore, I suggest that the use of ECMO during post-caridac arrest care cannot free from a contamination issue for prognostication by using biomarker (S100B and/or NSE). This possible limitation should be included in this submission.
  4. Authors can consider expanding upon what they think can build upon this prediction scheme. What other predictors for BD in OHCA survivors we should consider to improve accuracy such as EEG? Do you think adding other predictors would help? If adding the additional analysis using other predictors is not available, please suggest the effective strategy for predicting BD in patients treated with post-cardiac arrest care using multi-modal approach.

Author Response

Response to Reviewer 1 Comments

Thank you for the opportunity to review this submission. As a standard practice, I have thoroughly read this submission before coming to this conclusion. It was interesting to predict brain death after post-cardiac arrest care. Although this manuscript is well-written, it has a few limitations that need to be addressed before its acceptance.

We would like to thank you for the insightful comments. We have reviewed your comments carefully and revised the content of the manuscript as suggested. We strongly believe that your recommendations have enhanced the quality of our manuscript.

Point 1: Interestingly, the median age of the BD group was 45 years which is significantly younger than that of the non-BD group. Is it because of socio-cultural differences?

Response 1: We believed that it is mainly caused by two reasons. First, young patients without comorbidity are more likely to recover from these non-brain injuries and to be diagnosed with brain death. Even in studies conducted in Western countries, patients with brain death (BD) were younger than without BD. The age significantly differed between patients with and without BD (Chest 2021, 160, 139–147) and young age was independently associated with BD (Ann Intensive Care 2019, 9, 45). Furthermore, Chea et al (J Crit Care 2016, 32, 63–67) also indicated that younger age was independently associated with the development of central diabetes insipidus, which is usually an early event preceding BD. As mentioned in second paragraph of discussion section, some elderly patients were substantively similar to BD patients but inevitably end up classified as non-BD because they suffered rearrest or multisystem organ failure. Second, we believe that the socio-cultural context in which brain death is diagnosed on the premise of organ transplantation is additional reason for this difference.

Point 2: The authors used an “eye-opening/response” as a tool for predicting BD. However, its definition was not clear. Please clarify whether it is spontaneous or reactive to voice or pain. It can lead a confusion to readers. Please clarify the definition of “eye opening/response”

Response 2: We totally agree with your concern. No eye opening was defined if patients did not open their eyes spontaneously. Since this study focused on the occurrence of BD, we used the poor degree of neurological examination results.

To clarify our methods, we modified methods section as follows.

--> In all participants, we evaluated whether pupillary light reflex (PLR) and spontaneous eye opening were bilaterally present or absent and the Glasgow coma scale (GCS) motor grade for 5 days.

Point 3: ECMO induced hemolysis is widely accepted as one of the major complication of one. Therefore, I suggest that the use of ECMO during post-caridac arrest care cannot free from a contamination issue for prognostication by using biomarker (S100B and/or NSE). This possible limitation should be included in this submission.

Response 3: Thank you for your critical comment. In methods section we already stated that the serum with significant hemolysis were discarded. However, the absence of detectable hemolysis on the subsequent samples may not completely rule out the absence of NSE and S100B originating from red blood cells especially in cases with ECMO. We added the following sentences to the limitation section as suggested.

--> Third, as hemolysis leads to falsely elevated NSE and S100B values, we discarded samples with visible hemolysis. However, the absence of detectable hemolysis may not completely rule out the absence of NSE and S100B from extracerebral origin, especially in patients with ECMO support.

Point 4: Authors can consider expanding upon what they think can build upon this prediction scheme. What other predictors for BD in OHCA survivors we should consider to improve accuracy such as EEG? Do you think adding other predictors would help? If adding the additional analysis using other predictors is not available, please suggest the effective strategy for predicting BD in patients treated with post-cardiac arrest care using multi-modal approach.

Response 4: With regard to your comments, we can only speak based on the results that are presented in this manuscript. However, we believe that an early determination of BD by clinical examination could lead to false-positive results (Crit Care Med 2011, 39, 1538-1542). Therefore, a cautious approach is warranted when using TTM, and confirmatory tests such as EEG should be considered, especially when available as an adjunct to clinical determinants. To facilitate the determination process for BD, we recommend that physicians use initial GWR results and a series of S100B levels for screening, paying attention to clinical examination.

Reviewer 2 Report

I am happy to provide my opinion on the work from Hwan Song , Sang Hoon Oh and Hye Rim Woo focusing on the brain death and its prediction in out-of-hospital cardiac arrest patients who were treated with targeted temperature management. This retrospective work investigates the clinical variables in the period after ROSC with goal of process evolution evaluation of patients with brain death.

I have few suggestions and questions to the authors:

Abstract: abstract is missing good English flow and should be revised. I recommend adaptation of abstract according to the 5 sections, introduction, methods, results, discussion and conclusions.

Introduction:

Sentences in the line 49 and 52 should be revised as the information is not clear. Furthermore, the meaning of “CPR guidelines” in this sentence is not clear.

 Material and methods

Postcardiac arrest care – please cite international recommendation for this TTM.

Were patients only sedated with midazolam, and there was no analgesia treatment? Was rocuronium used in all patients, and what was the need for use of rocuronium?

How did you evaluate the GCS motor response if the patients were treated with rocuronium?

Authors should be congratulated for great work!

Author Response

Response to Reviewer 2 Comments

I am happy to provide my opinion on the work from Hwan Song , Sang Hoon Oh and Hye Rim Woo focusing on the brain death and its prediction in out-of-hospital cardiac arrest patients who were treated with targeted temperature management. This retrospective work investigates the clinical variables in the period after ROSC with goal of process evolution evaluation of patients with brain death.

We appreciate your excellent critique and helpful suggestions for our manuscript. We have reviewed your comments carefully, responded to the comments in a point-by-point manner, and revised our manuscript accordingly. We believe that it has enhanced the quality of the manuscript.

I have few suggestions and questions to the authors:

Point 1: Abstract

Abstract is missing good English flow and should be revised. I recommend adaptation of abstract according to the 5 sections, introduction, methods, results, discussion and conclusions.

Response 1: We strongly agree with you and prefer to follow the style of structured abstract. However, unfortunately, we’ve found that “Diagnostics” permits single paragraph of 200 words maximum without headings as abstract. In revised manuscript, we just added the word “in conclusion’ within the word count limitation suggested by the submission guideline for authors.

Point 2: Introduction:

Sentences in the line 49 and 52 should be revised as the information is not clear. Furthermore, the meaning of “CPR guidelines” in this sentence is not clear.

Response 2: We agree with your concern. In revised manuscript, we modified the sentences and added the reference numbers as follows.

--> Previous studies have reported some early factors related to the evolution toward BD [16-19]. However, it is unclear whether the prognostication tools recommended by international guideline for post-resuscitation care to predict neurological outcome can also predict the evolution toward BD [14]. Moreover, this analysis should be conducted in non-survivors.

Point 3: Material and methods

Postcardiac arrest care – please cite international recommendation for this TTM.

Response 3: In revised manuscript, we cited international recommendations for this TTM as suggested.

--> 2.2. Postcardiac arrest care

During the study period, all adult OHCA patients who were comatose after ROSC were eligible for TTM, usually at 33°C for 24 h [14].

Point 4: Were patients only sedated with midazolam, and there was no analgesia treatment? Was rocuronium used in all patients, and what was the need for use of rocuronium?

Response 4: We routinely used midazolam and rocuronium in all patients and no analgesia. Recent observational study reported that a continuous infusion of neuromuscular blocker (NMB) was associated with increased hospital survival and had beneficial effects on intensive care unit survival in cardiac arrest patients treated with TTM [Resuscitation 2013; 84:1728–1733. Resuscitation 2014; 85:1257–1262]. Improved tissue perfusion and decreased metabolic demand were proposed as possible mechanisms involved in the decreased lactate levels after a continuous infusion of NMB.

Point 5: How did you evaluate the GCS motor response if the patients were treated with rocuronium?

Response 5: We simply evaluated the GCS motor response to painful stimulus regardless of use of rocuronium. Rocuronium was used for TTM in all patients and discontinued as soon as the central temperature reached 35°C (usually on day 2). Therefore, regardless of whether there was a brain injury, the motor responses were suppressed on day 1, and in generally, patients could recover motor response on day 2 or day 3.

Authors should be congratulated for great work!

We thank you again for your advice on improving the quality of the manuscript.

Round 2

Reviewer 1 Report

The authors have fully responded all my comments and queries.

I have no more additional comments for the authors.